# Influence of Price Level and Perceived Price Dispersion on Consumer Information Search Behaviour: Moderating Effect of Durables and Consumables

**Weiqing Li [1], Qianyi Dan [2], Maomao Chi [1],\* and Weijun Wang [3],\***

1 School of Information Management, Central China Normal University, Wuhan 430079, China; liwq@mails.ccnu.edu.cn
2 School of Computer, Central China Normal University, Wuhan 430079, China; danqy@mails.ccnu.edu.cn
3 Key Laboratory of Adolescent Cyberpsychology and Behavior, Ministry of Education, Central China Normal University, Wuhan 430079, China
\* Correspondence: chimaomao@aliyun.com (M.C.); wangwj@mail.ccnu.edu.cn (W.W.)

**Abstract:** The methods consumers use to reduce their perceived risk and make reasonable purchase decisions can be synthesized under the umbrella term "consumer information search behavior" (CISB). As one key factor that conveys a product's value and quality, price has a significant impact on CISB. There are few studies that comprehensively consider the impact of price level (PL) and perceived price dispersion (PPD) on CISB, and there is a certain disagreement about the impact of PPD specific to the online shopping environment. To address this research gap, we construct a model using the data from 5515 consumers' purchasing and browsing behavior on a B2C e-commerce website, selecting six products as our research objects. We use a hierarchical regression analysis method to study the influence of product PL and PPD on CISB, and to explore the moderating effect of product categories (durables and consumables) on the relationship between PL, PPD and CISB. The results show that PL significantly affects CISB, and that product categories have a significant moderating effect on the relationship between PL and CISB. For durable goods, when the PL is high, consumers tend to increase their search behavior, both in depth and in breadth, and for consumables with low PL but higher purchase frequency, consumers likewise tend to increase their search behavior. In the B2C online shopping environment, PPD has a significant positive effect on CISB, and product category has a moderating effect on the relationship between PPD and CISB. When consumers purchase consumables, the higher the PPD, the higher the depth of CISB. The findings have several implications for marketing practitioners and enterprises advertising, also can help customers save time and energy in their search behaviors.

**Keywords:** consumer information search behavior; price level; perceived price dispersion; durables and consumables; moderating effect

## 1. Introduction

With the rapid development of the Internet and e-commerce, online consumption is becoming more and more popular and convenient. Consumers' online behaviors and preferences have greatly influenced the decision-making and development of enterprises and e-commerce platforms. Consumer behavior has always been a research hotspot, both in academia and industry. The buying process is typically affected by many factors; the selection of subsequent decision to purchase is based on a consumer's cognitive ability to fully evaluate alternatives and includes five stages: demand confirmation, information search, program evaluation, purchase decision and post-purchase evaluation. Consumers make purchase decisions about products in a similar way to the process of answering questions: first, they gather information about products to form a cognition about product-related function, brand and value. Then, they carefully weigh the pros and cons, finally,

they make a decision [1]. And to deal with the proliferation of options, consumers shopping online will either use price search engines or compare prices at e-commerce platforms [2]. A full information search for pre-ordered products is an important basis for consumers to make reasonable purchase decisions; understanding CISB is of great significance to the formulation of corporate strategy and product positioning.

Previous studies have confirmed that risk aversion is the root cause of CISB [3,4], to reduce uncertainty and risk, consumers usually try to search relevant product information before buying [5], and as an important factor in delivering product quality and value, a product's price significantly affects consumers' perception of risk and plays a crucial role in their purchasing behaviors [6,7]. Scholars have conducted a large number of studies on the impact of price factor, which is usually the most important influencing factor for search behavior. To enjoy the maximum utility from their purchase and reduce perceived financial risk, consumers usually spend more of the total research time searching specifically for product price information [8,9]. When the product price is high or rising, buyers may conduct a more thorough search, a phenomenon closely related to the product's price and the benefits that a consumer may derive from its purchase [10]. In addition, the perception of price dispersion also has a significant impact on CISB. Generally, there is a positive correlation between PPD and CISB; if consumers feel that the price and quality of a product to be different, they will conduct a more extensive search for information before making the purchase [11–13]. However, some studies suggest that, in the online shopping environment, when the PPD is high, consumers tend to prefer the sellers with whom they are familiar because of a distrust of unknown sellers, therefore, higher PPD may actually lead to a reduction in search effort online [14,15].

Existing research on the effect of product price factors on CISB has the following limitations: (1) Scholars usually study the effects of PL and PPD on CISB separately [11,16,17]. (2) Owing to the differences between the purchase situation (online or offline) and the e-commerce platform (C2C or B2C), use the methods of questionnaires and experimental, may be too limited to accurately describe the consumers' PPD, so there is a certain disagreement about the impact of PPD on CISB specific to the online shopping environment in such studies [11,12,14,15]. To fill this void, this study uses data about consumer online purchases and browsing behaviors from B2C e-commerce websites to measure the perceived PL and PPD as well as the browsing time, times and types of products during each purchase process. To identify patterns of perceived risk, we use a multiple regression analysis method to study the impact and motivation about perceived PL and PPD on the breadth and depth of CISB. We selected refrigerators, electric kettles and U disks as the research objects of durable goods and infant milk powder, facial cleanser and tissue as the research objects of consumable goods to study the moderating effect of product categories. The main innovations and discoveries of our work include the following: (1) Using actual consumer network behavior data, we conducted a comprehensive analysis of the effects of product PL and PPD on CISB, and tested the moderating effects of product category. (2) We found that the effect of PL on CISB is neither linear nor directly proportional, when the price of the product reaches a certain level, consumers tend to conduct a more extensive search because of they may find more suitable or cost-effective products. Consumers will also conduct more research for products with a lower price but high purchase frequency. (3) In the B2C online shopping environment, PPD was found to have a positive impact on CISB, and product category has a significant moderating effect between consumers' PPD and CISB.

This paper is organized as follows: in Section 2, we review the literature on the influence of PL and PPD on CISB, and then propose the research hypotheses. Section 3 describes the data and research methods. Section 4 discusses the results we found by data analysis, while Section 5 presents our conclusions, implications and future direction.

## 2. Literature review and Research Hypotheses

### 2.1. Literature Review

Consumers' purchase decisions depend heavily on consumers processing the results of the search for product information. More specifically, the term "information search behavior" is defined as "actively activating from memory knowledge or obtaining relevant information from external environment" and is divided into two search parameters: internal and external. An internal search refers to the process of a consciously activating relevant knowledge in one's memory, whereas an external search refers to the behavior of an individual to obtain relevant information from the outside world [18]. In general, the information search process starts with an internal search. According to this process, a consumer first recalls previous experiences and internal evaluations of related products. If a consumer thinks that their memories and experiences can fully evaluate a related product, they will discontinue their search for further information. When the experience is insufficient, however, or the memory conflicts with some new information, the consumer will search for information from the outside in order to eliminate arising conflicts in perception and, as a result, reduce any perceived risk that might come from the purchase. The external search is more proactive than its internal counterpart, and actually requires a certain amount of time and money to complete. This initiative reflects the tendency of consumers to search for information and eventually go down a decision-making path, which is generally geared toward obtaining greater consumption benefits [19]. The external search behavior can be further divided into the breadth and depth of search, in which depth represents the time and quantity of relevant product information consumers obtain, while breadth signifies the type or types of product information that consumers obtain through their search efforts [20].

Consumers' tendency to avoid risk will positively affect their search behaviors [3,4]. An important underlying cause of risk perception is uncertainty due to a lack of information or experience; the most effective way to reduce this uncertainty is to extensively search for relevant product information to bridge the knowledge or experience gap [21]. Murray [22] believes that the magnitude of perceived risk helps determining a consumer's need for information, and that consumers will search for information sources, information types and a quantity that can best meet their needs to reduce uncertainty before making a purchase.

Several factors can affect a consumer's effort in the search for information, including those that are behavioral, personal, product- or service-related and those that are situational [23,24]. And this study focuses on the impact of the product price factor on CISB. Our first assumption is that consumers will learn as much as possible about a product in order to make a rational choice before making the purchase, and that the differences between consumers are negligible. Second, we assume price to be one of the most important indicators of a product's benefit to the consumer. Because price is used to convey the quality and value of products to consumers, it can be observed to be part of just about every possible transaction. Therefore, we assume for this study that price factors play an important role in a consumer's purchasing decisions [8,9]. Additionally, a consumer's purchasing decisions not only depend on the price of products, as they are set by individual stores, but also from a psychological expectation of price, which comes about through a comparison of the prices of various homogeneous products [25]. Therefore, the influence of price factors on a consumer's purchasing decisions has the following framework: (1) If a consumer believes that a store's products are relatively cost-effective, they will buy there. (2) If a consumer thinks the price is relatively expensive, as long as the search cost is not too high, they will search other stores to find homogenous products with lower prices [26]. For consumers, the perceived price is more important than the actual price.

Therefore, the product PL and PPD will affect CISB. Price dispersion refers to the degree of deviation from a particular center of the price distribution of homogeneous products. Homogenous products refer to products of the same brand or type, as well as those with the same functions. Product price dispersion arises, in part, due to the heterogeneity of sellers, and partly due to the imbalance of available consumer information [27,28].

When prices are available to consumers before purchasing, CISB is usually guided by price [29]. Wolinsky [30] proposed a consumer information search model, pointing out that consumers will learn the price of the product through a large number of searches, and find the products with the best perceived value among horizontally differentiated sellers. Urbany et al. [31] proposed and validated the model of consumer price search determinants in the retail market, whereby consumer price search behavior is defined as the cost of acquiring and comparing competitive prices. This model indicates that product PL and PPD positively affect consumer price search behavior. Biswas et al. [11] pointed out that large discrepancies between prices at different locations make it difficult for consumers to accurately judge a product's value and therefore will increase the uncertainty of their purchase decision, resulting in an increase in the consumer's perception of risk. In the offline shopping environment, PPD has a positive effect on CISB: if consumers feel that the price and quality of products in the market are more different, they will conduct a more extensive search for information before buying, because when the product's price difference is greater, consumers will be motivated to search more to reduce their perceived financial risk and obtain the most benefit from their purchase [11,12]. Other studies have pointed out that, in the online shopping environment, when the PPD is high, consumers tend to prefer the online retailer they are familiar with than to search in other stores, so a higher degree of price dispersion may lead to a reduction in online searching [14,15].

In summary, CISB is the basis of consumers' purchase decision-making framework. It is the process of inquiring and obtaining product information in external environments for processing and making reasonable purchase decisions; the fundamental for this behavior is to reduce the uncertainty of decision-making to avoid risk, and to maximize the benefit to the purchase decision. As a factor affecting a consumer's decision-making, price is usually one of the most important factors that affect consumers' decision-making behaviors. However, the studies on the influence of product price on CISB are still limited; the influence of PL and PPD on purchasing decisions are two topics usually studied and discussed separately. And in the online shopping environment, the influence of PPD on CISB exists a great divergence. Based on these specific limitations of the previous research, we developed the following hypotheses in Section 2.2.

*2.2. Research Hypotheses*

2.2.1. The Effect of PL and PPD on CISB

According to perceived value theory, a product's price needs to be consistent with its perceived value. If the price consumers expect to pay is different from what they previously paid, they must decide whether this difference is important enough to them to take corresponding action [32]. Information economics theory suggests that the greater the economic benefits consumers perceive during the purchase process, the more extensive their search for price information they conduct will be [11]. Therefore, when purchasing high-value products, consumers usually spend more time searching for product information in order to reduce perceived risk and obtain higher expected returns [33]. Consumers have been found to be insensitive to the costs of searching for information when purchasing expensive products, meaning they tend to disregard information search costs; product prices have been found to be positively correlated with the amount of search time and number of searches for information [34]. The higher the price of the product, the higher the consumer's perceived price and the greater the perceived cost. When a consumer purchases a product that they are unsatisfied with and which has high price, he or she may face a higher financial risk and a greater loss. Therefore, as the price of a product goes up, so too does the consumer's willingness to spend time searching for information about it.

In addition, a high level of PPD increases the price sensitivity of buyers and the uncertainty of transactions [35]. Buyers may be skeptical of unusually low- or high-priced products. The high level of PPD brings consumers more uncertainty and increases the perceived risk of such transactions [36]. A perception of high price dispersion also encourages buyers to search more and assess the differences between alternatives in order

to find the most suitable product. Therefore, the PPD has a positive impact on a consumer's perceived risk, which in turn affects their search behaviors [37]. Based on these criteria, we propose the following hypotheses:

**H1a:** *In the online shopping environment, product PL positively affects CISB.*

**H2a:** *In the online shopping environment, PPD positively affects CISB.*

2.2.2. The Effect of Product Type on CISB

For different types of products, consumers will have different ways of acquiring and screening information to make purchasing decisions [38]. Consumers also have different perceptions of risk according to different product categories as well. In essence, when faced with higher perceived risk, a consumer will search for more information to reduce risk; they will pay more attention to the reliability of information and hope to make reasonable judgments with the help of that information [39,40]. Therefore, when consumers buy more expensive, more conspicuous, less familiar or more complex products, they tend to conduct more information search. Product category and familiarity also directly affect CISB. For instance, Biswas et al. [11] found that, while shopping online, consumers have a higher perceived risk for non-digital products (i.e., clothing, cookware) and tend to invest more in the search for information. Wu et al. [37] found that when purchasing high-touch products (product attributes are difficult to perceive and judge), the positive impact of PPD on perceived risk is greater than the purchase of low-touch products (product attributes are relatively uniform, easy to understand and access). Muller et al. [41] found that when purchasing durable goods, consumers tend to invest more cognitive resources in product selection process and conduct more comparisons and evaluations before purchasing. We found that for products with a higher price and more complex functional attributes, consumers usually conduct a more extensive search for information. Compared with consumables, durable goods usually have higher price, longer life cycle, more emotional input and after-sales service involved, and may face higher perceived risk. Therefore, consumers tend to invest more cognitive resources and conduct a more substantial comparison and evaluation in the process of selecting a product [37,42]. Based on these factors, we propose the following hypotheses:

**H1b:** *Product category (durable or consumables) has a moderating effect between PL and CISB. When purchasing durable goods, product PL has a greater impact on CISB.*

**H2b:** *Product category (durable or consumables) has a moderating effect between PPD and CISB. When purchasing durable goods, PPD has a greater impact on CISB.*

The summary of the theoretical model presented in this study is shown below in Figure 1.

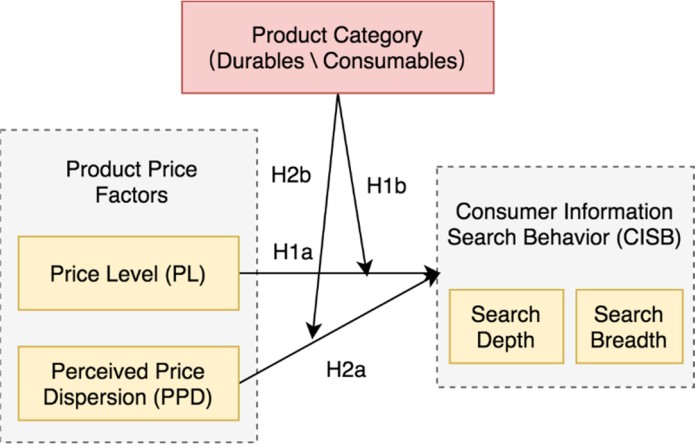

**Figure 1.** The proposed research model.

## 3. Research Data and Methods

This section describes our proposed approach, starting with the intuition behind it, and then proceed to describe it in detail. Clickstream data record the entire search path of consumers and is a reliable source of information about consumer behavior. By analyzing consumer purchasing and browsing behaviors, we can excavate consumers' preferences. We use historical data of the Suning B2C e-commerce website, including 37,159 consumers' purchasing behaviors data and 381,889 pieces of product information data. There are 5515 consumers who purchased the six products selected for this study; all data do not contain privacy information the data format of each consumer behavior is as follows, and the meaning of each field is shown in Table 1.

**Table 1.** Consumer behavior data fields and explanations.

| Data Type | Examples | Meaning |
| --- | --- | --- |
| User ID | 0000066c84079975340f9ed898825551 | User unique identification code |
| | BROWSE | Browsing behavior |
| | CART | Add to cart |
| User behavior | COLLECT | Collect |
| | BUY | Buy |
| | OTHER | Other behavior |
| Product ID | 000000000126823313 | Product unique identification code |
| UNIX timestamp | 1509670166000 | Time when the behavior occurred |

Product information data include product number, product name, product image URL, product price, product URL, first-level category, second-level category, third-level category and brand name. We use multiple regression analysis, but the average price and actual price dispersion of each product are used as classification variables. Therefore, the independent variable of this study was calculated by extracting product price information and browsing the behavior data of each consumer.

In practice, consumers can only evaluate a handful of products, in actual network transaction data, incorporating this requires a new latent variable, the consumers' consideration set; that is, the set of products the consumer actually chooses between [43]. So We assume that consumers browse $n$ such products each time they purchase product C, the prices are $p_1$, $p_1$, $p_1$ ... ... $p_n$, and $p_n > p_{n-1} > ... > p_1$, and we defined the perceived PL of each consumer as

$$PL = \sum_{i=1}^{n} p_i \frac{tf_{(i)}}{\sum_{j=1}^{n} tf_{(j)}},$$ (1)

where $\frac{tf_{(i)}}{\sum_{j=1}^{n} tf_{(j)}}$ represents the frequency of users' attention to the product.

In empirical studies, price variance, standard deviation or dispersion (the magnitude of price changes around the mean) is often adopted to measure price dispersion. In this study, the standard deviation of the price of all the products browsed by the consumer is selected as the degree of price dispersion of the consumer during the purchase of this product. We use the standard deviation of the price of all products browsed by consumers as the PPD in the process of purchasing this product,

$$PPD = \sqrt{\frac{\sum_{i=1}^{n} (p_i - PL)^2}{n - 1}}$$ (2)

Then, we extract the consumer behavior information for all six products purchased, taking each consumer as a unit. Next, we calculate the browsing behaviors of this consumer when purchasing such products, according to the definition of the depth and breadth of CISB. This article defines the total number of times and total time consumers browse a product as the depth of information search, and the breadth of information search is

defined as the total variety of products that consumers browse [44]. Thus, the dependent variables defined in this study are, "BROWSETIMES" refers to the number of views for each purchase; "BROWSETIME" refers to the browsing time for each purchase; "BROWSETYPE" refers to the types of browsing for each purchase.

Finally, multiple regression analysis is used to explore the impact of PL and PPD on CISB and study the moderating effect of the product category.

## 4. Data Analysis and Results

### 4.1. Research Objects and Data Description

Firstly, we selected five kinds of durable goods (Refrigerator, laptop, electric rice cooker, Electric kettle, U disk) and five kinds of consumables (Infant milk powder, rice, Facial cleanser, toothpaste, Tissue) according to the definition of durable goods and consumables in literature [38], then we found out all the consumers who have purchased the above 10 products from the dataset. And finally, we chose Refrigerator, Electric kettle, U disk, Infant milk powder, Facial cleanser, Tissue as research objects because the sample size of the other four products is small. Descriptive statistics of price factors and consumer browsing behaviors for each product are shown in Table 2.

**Table 2.** Research objects and descriptive statistics of variables.

| Products | Minimum Price | Maximum Price | Average Price | Standard Deviation of Price | BROWSETIMES | BROWSE TIME | BROWSE TYPE |
|---|---|---|---|---|---|---|---|
| Refrigerator | 399 | 29,052 | 3408.33 | 72.51 | 135.81 | 178.42 | 6.64 |
| Electric kettle | 31 | 1680 | 298.24 | 22.96 | 32.53 | 50.08 | 2.29 |
| U disk | 15 | 1269 | 106.65 | 20.75 | 18.26 | 47.36 | 1.20 |
| Infant milk powder | 74 | 2568 | 270.31 | 18.71 | 35.99 | 95.27 | 2.51 |
| Facial cleanser | 19 | 599 | 65.66 | 8.99 | 52.53 | 68.47 | 3.29 |
| Tissue | 14 | 233 | 34.92 | 6.51 | 83.29 | 131.14 | 4.64 |

### 4.2. Data Analysis

The correlation coefficient analysis matrix of each variable is shown in Table 3. The PL and the PPD are significantly positively correlated with CISB.

**Table 3.** Correlation analysis.

| Variables | 1 | 2 | 3 | 4 | 5 |
|---|---|---|---|---|---|
| 1, PL | 1 | | | | |
| 2, PPD | 0.719 ** | 1 | | | |
| 3, BROWSETIME | 0.163 ** | 0.189 ** | 1 | | |
| 4, BROWSETIMES | 0.184 ** | 0.206 ** | 0.365 ** | 1 | |
| 5, BROWSETYPE | 0.198 ** | 0.222 ** | 0.454 ** | 0.788 ** | 1 |
| Mean | 1530.83 | 27.27 | 179.65 | 89.72 | 4.82 |
| Variance | 2580.55 | 33.65 | 197.15 | 136.22 | 4.91 |

Note: ** significant at level 0.01.

Taking PL and PPD as independent variables, BROWSETIME, BROWSETIMES and BROWSETYPE as dependent variables and product category as moderate variable, and using SPSS analysis tools, the results of multiple hierarchical regression analysis of CISB are shown in Table 4.

**Table 4.** Multiple hierarchical regression analysis.

| Independent Variable | BROWSETIME | | BROWSETIMES | | BROWSETYPE | |
|---|---|---|---|---|---|---|
| | M1 | M2 | M3 | M4 | M5 | M6 |
| PL | −0.104 ** | −0.075 * | −0.095 * | −0.075 * | −0.119 * | −0.098 * |
| PPD | 0.202 ** | 0.15 * | 0.156 ** | 0.110 ** | 0.172 ** | 0.145 * |
| Interaction effect | | | | | | |
| Product Category × PL | | −0.115 * | | −0.048 * | | −0.068 * |
| Product Category × PPD | | 0.207 ** | | 0.166 ** | | 0.055 |
| R2 | 0.033 | 0.053 | 0.048 | 0.057 | 0.053 | 0.144 |
| F | 95.03 ** | 57.804 ** | 139.39 ** | 24.919 ** | 155.68 ** | 47.93 ** |
| ΔR2 | 0.033 ** | 0.020 ** | 0.048 ** | 0.009 ** | 0.053 ** | 0.016 ** |
| MAX_VIF | 1.051 | 3.724 | 1.236 | 2.89 | 1.157 | 5.907 |

Note: * significant at the 0.05 level, ** significant at the 0.01 level.

To verify hypothesis H1a and H2a, we first decentralized the variables, then we performed the regression analysis, taking PL and PPD as independent variables and BROWSETIME, BROWSETIMES and BROWSETYPE as dependent variables. The results are shown in Table 4. First, variance inflation factors (VIFs) are all less than 10, indicating that the multicollinearity had no significant effect. Regression analysis results of models M1, M3 and M5 show *PL* to have a significant and negative influence on search time ($M1$, $\beta = -0.104$, $p < 0.01$), number of searches ($M3$, $\beta = -0.095$, $p < 0.01$) and search types ($M5$, $\beta = -0.119$, $p < 0.01$). PPD had a significant and positive influence on search time (M1, β = 0.202, p<0.01), number of searches ($M3$, $\beta = 0.156$, $p < 0.01$) and search type ($M5$, $\beta = 0.172$, $p < 0.05$). These results support the confirmation of hypothesis H2a.

To verify hypotheses H1b and H2b, models M2, M4 and M6 include interactive terms to perform the regression analysis testing the moderating effect of product category. The results show both ΔR2 and ΔF to be significant, indicating the models' fit and explanatory power were improved. Additionally, product category was found to have a significant moderating effect between *PL* and CISB ($M2$, $\beta = -0.115$, $p < 0.05$), ($M4$, $\beta = -0.048$, $p < 0.05$), ($M6$, $\beta = -0.068$, $p < 0.05$). Product category also had a significant moderating effect between *PPD* and consumers' depth of search ($M2$, $\beta = 0.207$, $p < 0.01$), ($M4$, $\beta = 0.166$, $p < 0.01$). These results support H1b and only partially support H2b.

The interaction effects of product category are shown in Figure 2a through Figure 2f below.

From Figure 2a–c, we found that higher PL for durable products tended to result in increased search behaviors, though among consumables, a higher PL resulted in reduced search behaviors, partially supporting hypothesis H1a. From Figure 2d,e, we found that PPD positively affects CISB, and also has a more significant effect on the depth of CISB for consumables.

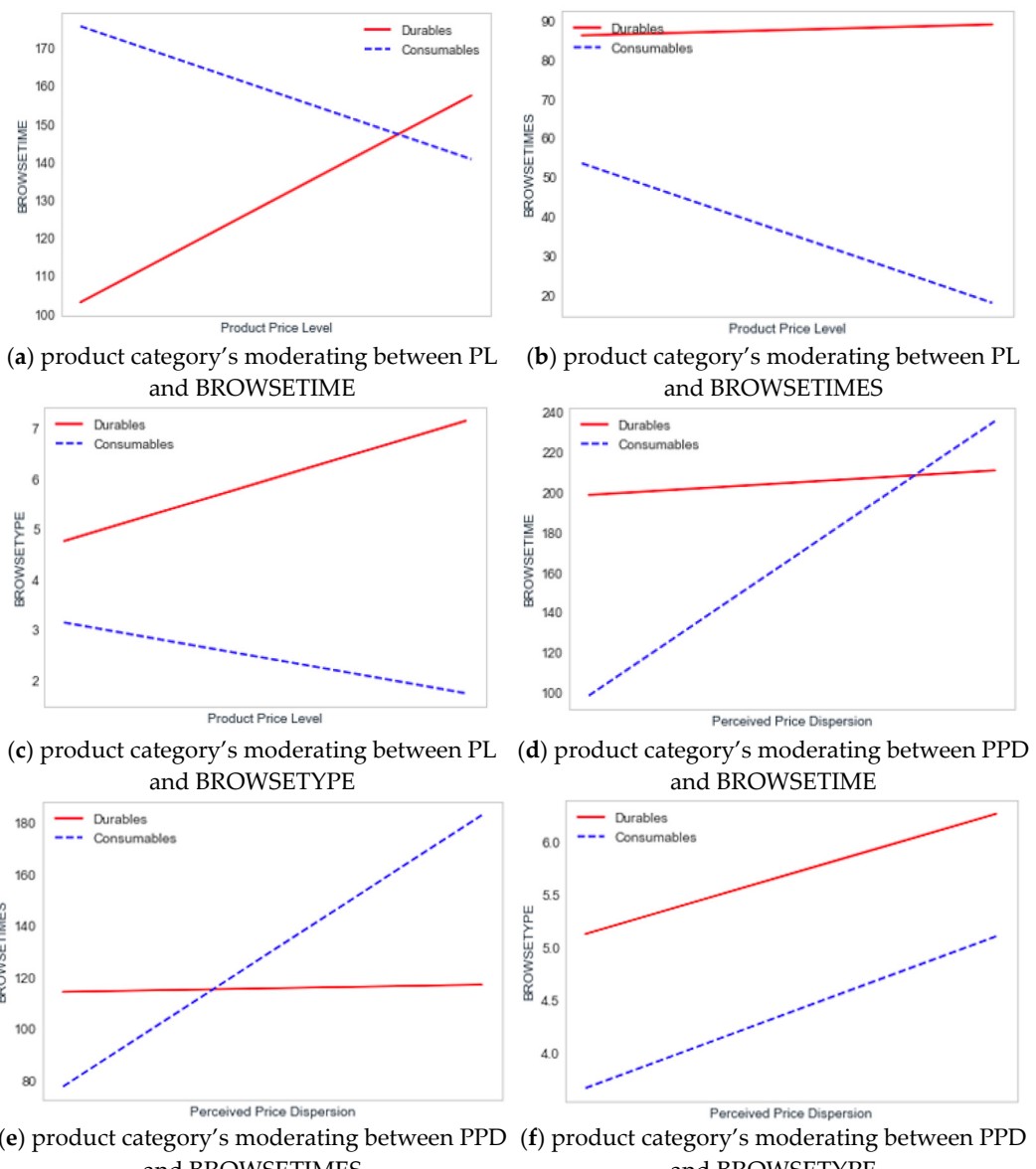

**Figure 2.** Interaction effects of product category.

## 5. Discussions

Analyzing users' network behaviors and identifying users' preferences are undoubtedly the key to improve the performance of enterprises and e-commerce platforms. This study was a research endeavor to unearth the role of product price in influence of the CISB, including PL and PPD. We provided experimental evidence from the consumers' behaviors on B2C e-commerce platforms regarding the PL and PPD on CISB. And our findings unearthed that the product category (durable and consumable) have a significant moderating effect on the relationship between PL, PPD and CISB.

Specifically, in the online shopping environment, the PL of durable goods has a significant positive impact on CISB. The more expensive the durable is, the higher the breadth and depth the consumer's search effort is predicted to be. Because consumers have a higher perceived risk for higher-value, longer-lasting durable goods, more information searching behavior is beneficial as a way to reduce consumers' perceived risk and to obtain higher economic benefits. Conversely, PL was found to have a negative impact on the CISB of consumable goods. Product category (durable and consumable) was also found to have a significant moderating effect on the relationship between PL and CISB. It offers

different conclusions from those put forward in existing literature [33,45], that consumers will invest more effort searching for information about high-priced products. In this study, we found that CISB increases with the increase of product PL for durable goods. However, for consumable goods, which are characterized by a lower price and a higher frequency of purchase, consumers were shown to search for more information to obtain higher economic benefits for their frequent purchase behaviors. These conclusions are notable because of their consistency with information economics theory.

Furthermore, In the online shopping environment, PPD has a significant positive impact on CISB, and the product category has a significant moderating effect on the relationship between PPD and CISB. When consumers purchase consumables, their PPD has a more significant impact on the depth of CISB. Because consumables are purchased more frequently, higher degrees of PPD encourage consumers to make more trade-offs and comparisons on the price, discounts and preferential information about the products in order to ascertain the highest benefit from making the purchase. Different from the conclusions in other previous studies [11,14,15], in the online shopping environment, customers faced with a high level of PPD tend to prefer trusted brands or stores, and give up searching for more information and price comparisons. We found that, in the B2C online shopping environment, PPD has a positive impact on CISB. Therefore, we infer that on the B2C platform, consumers' trust for retailers has improved, and they pay more attention to the cost performance of homogeneous products.

The third point is an inference. It can be found from the descriptive statistics of product data in Table 2, the depth and breadth of the search for information about refrigerators and tissues are much higher than for other products. We can easily understand that the PL of refrigerators is much higher than other products, but the phenomenon of tissues is unclear. We inspected the original consumer behavior data and found that consumers purchased tissues much more frequently than other products. So we speculate that if consumers need to buy certain consumables frequently, their search behavior for this product will be high [46].

## 6. Conclusions

This study answered how PL and PPD affect CISB synthetically. And we determined the impact direction of PPD on CISB in B2C e-commerce platform. In general, the research objectives were satisfactorily obtained, and our results have following implications and limitations.

### 6.1. Theoretical Implications

The results of study included some important theoretical implications in that, we prove that the PL and PPD of products affect the CISB at the same time, the PL affects the CISB through the degree of perceived risk, as more information search behavior can reduce the perceived risk [10]; and the positive impact of price dispersion on CISB is mainly due to the perceived potential benefits of more information search behavior [11–13]. In addition, our findings proved that consumers have different purchase decision-making thinking and paths for different categories of products from the perspective of consumers' information search behavior before purchase [38]. What is more, our research provides a new idea for user behavior research based on large-scale e-commerce data.

### 6.2. Management Implications

This study has several important implications for marketing practice as well, by providing new insights into CISB for products of different PL and PPD. Our findings suggest that the characteristics of durable goods, which tend to have a higher PL, or consumable goods, or those which are purchased with higher frequency, encourage consumers to search for more information about these products. Therefore, we recommend marketing practitioners to provide consumers with abundant and varying product information to satisfy their need for such comprehensive levels of pre-purchase information. We also

found that CISB is relatively lower for products characterized by moderate prices or a low PPD. Therefore, we recommend enterprises selling these goods to increase the intensity of advertising while ensuring consistently high product quality, to keep consumers who need to buy such products from looking to external sources for more knowledge about competing products. This strategy can help customers save time and energy in their search for regular use products, and help corresponding enterprises to sell more of their products, and faster.

### 6.3. Limitations and Future Directions

In view of the limitations of network data characteristics and research methods, this article has the following limitations: (1) CISB is affected by many factors, such as individual product knowledge, search attitude and time pressure, the number of alternatives, brand differences, complexity and symbolism of the product [17]. This study only considers the influence of price as a factor on CISB. (2) This study adopts an empirical research method; based on information economics theory, the research results can be supplemented and improved by combining consumer information search models, such as optimal stop model, sequential search model [47], fixed search model [48], or others. (3) This study proved price factors significantly affect consumers' search behavior before purchase and purchase decision, and future research could examine the association between price-sensitivity and consumer characteristics which can help to make targeted price promotions and effective personalized recommendation [49].

**Author Contributions:** Writing—original draft preparation, W.L.; data analysis and visualization, Q.D. and W.L.; writing—review and editing, M.C. and W.W.; funding acquisition, W.W. All authors have read and agreed to the published version of the manuscript.

**Funding:** This research is supported by Program of National Natural Science Foundation of China (No. 71571084, No. 71271099) and China Scholarship Council.

**Institutional Review Board Statement:** Not applicable.

**Informed Consent Statement:** Not applicable.

**Data Availability Statement:** The data used to support the findings of this study have been deposited in the Baidu Netdisk repository (https://pan.baidu.com/s/1PtG9j42EvMq8-7-aOdaKVw, password: 9ble).

**Conflicts of Interest:** The authors declare no conflict of interest.

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
