# Peer review of "Influence of Price Level and Perceived Price Dispersion on Consumer Information Search Behaviour: Moderating Effect of Durables and Consumables"

_sustainability, doi:10.3390/su13042105_

Round 1

Reviewer 1 Report

This study attempts to establish a link between price line and CISB and found that the price line has a positive and significant impact on CISB for durable goods. In contrast, the price line has a negative relation with consumable goods. The authors further argued that CISB increases with the price line for a durable good. This is an exciting study because e-commerce and online shopping have been gaining momentum over the years.  This manuscript is through and detailed study. I don’t have major comments on it.  However, it is not free from some grammatical errors. The writing can be improved substantially—further, there some repeated sentences throughout the text. Please attention to them. Please check the format and maintain consistency.

Comments:

  1. Page 1, line 5: Remove “,”  in Wang 2,*
  2. Page 1, line 6: “;” after China
  3. Page 1, line 9; “;” after China
  4. Page 1, line 18-21, a sentence is so long and unclear; please rephrase it.

Table 3, please use different symbols before PL and PDD as * indicates significance levels. Also, please format the table to make sure it stands alone.

Table 3 is repeated.

Author Response

Response to Reviewer 1 Comments

Point 1: This study attempts to establish a link between price line and CISB and found that the price line has a positive and significant impact on CISB for durable goods. In contrast, the price line has a negative relation with consumable goods. The authors further argued that CISB increases with the price line for a durable good. This is an exciting study because e-commerce and online shopping have been gaining momentum over the years.  This manuscript is through and detailed study. I don’t have major comments on it.  However, it is not free from some grammatical errors. The writing can be improved substantially—further, there some repeated sentences throughout the text.Please attention to them. Please check the format and maintain consistency.

Response 1:

       Thank you for your affirmation and pointing out the problems in the article. We have made following modifications according to your suggestion.

       1) We deleted the sentence “and that the main purpose of this behaviour is to reduce their perception of risk” in line 60-61 to make it concise.

       2) We deleted the sentence “The search for information is an important step in consumers’ purchase process and the basis for fostering satisfactory purchase decisions.” in line 114-115, because the problem of repeated terms.

       3) We rephrased the sentence “The fundamental cause of CISB has been argued to be risk aversion, a consumer’s tendency to avoid risk will positively affect their search behaviour.” in line 136-137 to “Consumers’ tendency to avoid risk will positively affect their search behaviours.”

       4) We checked the singular and plural usage of "behaviour" in detail, and rewrite "consumer search behavior" as CISB to make the article more concise.

Point 2:

  1. Page 1, line 5: Remove “,”  in Wang 2,*
  2. Page 1, line 6: “;” after China
  3. Page 1, line 9; “;” after China

Response 2:

       Thank you for your careful inspection and the above three valuable opinions, but after we looked at the new articles in this magazine, as figure 1, we found that the “,” in “Wang 2,*” and the “;” after China in line 6 are necessary, and we remove “;” after China in line 9. And thank you again for your conscientiousness and responsibility.

Figure 1 Title of Sustainability

Point 3:

Page 1, line 18-21, a sentence is so long and unclear; please rephrase it.

Response 3:

       Thank you for your valuable advice. So we rephrase the sentence “We use a multiple regression analysis method to study the influence of product PL and PPD on CISB in consumer purchase decision-making process and explore the moderating effect of product categories (durables and consumables) in this process.” to “We use a hierarchical regression analysis method to study the influence of product PL and PPD on CISB, and to explore the moderating effect of product categories (durables and consumables) on the relationship between PL, PPD and CISB.”

Point 4:

Table 3, please use different symbols before PL and PDD as * indicates significance levels. Also, please format the table to make sure it stands alone. Table 3 is repeated.

Response 4:

       Thank you for your valuable advice. First we change the second Table 3 to Table 4. And change the “*” before PL and PDD to “”.

Reviewer 2 Report

The paper is well structured and approaches an important topic

The structure of the paper is robust

The references are up-to-date

The analysis is refined

Some minor rewies of the English languages are needed.

Author Response

Response to Reviewer 2 Comments

The paper is well structured and approaches an important topic.

The structure of the paper is robust.

The references are up-to-date.

The analysis is refined.

Some minor rewies of the English languages are needed.

Response:

    Thank you very much for your high evaluation of our research.

    We have checked the manuscript carefully and made some linguistic and grammatical changes to make the article concise and rigorous.

    And we will continue to work hard to improve our work.

Reviewer 3 Report

It was a pleasure to read and review this very interesting paper of high scholarly standard. The paper explores the important topic of modern consumer behaviour, as online shopping has become perhaps the most chosen channel for purchasing goods during the COVID pandemic. The authors completely ignore this fact, and I absolutely support this approach, especially since many papers treat references to pandemics in a 'marketing' way. Or perhaps the clue to unclear consumer behaviour in the case of tissue purchases is precisely the time period (pandemic?) from which the online platform browsing data comes?

The methodology serves the purpose and hypotheses of the research well. Methodology and results are well and clearly explained. I would just suggest clarifying:

1 - what was the criterion for product selection, since in the case of frequent purchase goods (consumables) this is a rather surprising choice. Was the first step a selection of products or a search for consumers buying any six products, the same ones divided into two categories?

2 - it is also not clear why in Figure 1 both arrows H1a and H2a refer only to 'search depth';

3 - in Table 2, in the columns I propose to write the numbers with the same precision (spaces after the dot), and in the explanation under the table leave only p value <0.01

The paper is well organized, but I would suggest improving the chapter structure a bit, starting with chapter 5, as follows:

5. Discussion - I propose to start this chapter with the text in L. 338-342 and integrate it with the text in L. 349-385 (consequently, subsection 5.1. will disappear).

6. Conclusions - I propose to begin with the text in L. 343-347 and then continue with 6.1. Theoretical implications, 6.2. Management imp., 6.3. Limitations & ... (you can also drop the separation of these subsections).

The paper is well written, I recommend it for publication after the authors have made these minor corrections 

Author Response

Response to Reviewer 3 Comments

Point 1: It was a pleasure to read and review this very interesting paper of high scholarly standard. The paper explores the important topic of modern consumer behaviour, as online shopping has become perhaps the most chosen channel for purchasing goods during the COVID pandemic. The authors completely ignore this fact, and I absolutely support this approach, especially since many papers treat references to pandemics in a 'marketing' way. Or perhaps the clue to unclear consumer behaviour in the case of tissue purchases is precisely the time period (pandemic?) from which the online platform browsing data comes?

Response 1:

       Thank you for the valuable views and suggestions, which is also the value and significance of our research. Our research provides a new idea for user behavior research based on large-scale e-commerce data and without user research. But unfortunately, we don't have data on consumers' online behavior during the epidemic, and We use the data of Suning e-commerce platform from 2017 to 2018 provided by our partners LEO. So we didn't particularly emphasize the significance of research for the pandemic. In addition, we think your opinions are very valuable. It is very meaningful to study the behavior of online consumers during the pandemic. We will continue to conduct relevant research if we can get the corresponding consumer data during the pandemic.

The methodology serves the purpose and hypotheses of the research well. Methodology and results are well and clearly explained. I would just suggest clarifying:

Point 2:  what was the criterion for product selection, since in the case of frequent purchase goods (consumables) this is a rather surprising choice. Was the first step a selection of products or a search for consumers buying any six products, the same ones divided into two categories?

Response 2:

       Thank you for your valuable advice witch point out the shortcomings of our article. The process of selecting research objects is as follows:

       Firstly, we selected five kinds of durable goods (Refrigerator, laptop, electric rice cooker, Electric kettle, U disk) and five kinds of consumables (Infant milk powder, rice, Facial cleanser, toothpaste, Tissue) according to the definition of durable goods and consumables in literature [42], then we found out all the consumers who have purchased the above 10 products from the dataset. And finally we chose Refrigerator, Electric kettle, U disk, Infant milk powder, Facial cleanser, Tissue as research objects because the sample size of the other four products is small.

       And we add the above description to the original text in Chapter 4.1

Point 3: it is also not clear why in Figure 1 both arrows H1a and H2a refer only to 'search depth';

Response 3:

       Thank you for your valuable advice. Arrows H1a and H2a refer to CISB in our hypotheses, but it was unclear in “Figure 1”. So we changed the figure of research model to Figure 2 as below to make it more clear.

Figure 2. New figure of research model.

Point 4:in Table 2, in the columns I propose to write the numbers with the same precision (spaces after the dot), and in the explanation under the table leave only p value <0.01.

Response 4:

       Thank you for your valuable advice. We have modified the numbers with the same precision (two decimal places) in Table 2. And leave only p value <0.01 in the explanation under the table 3.

Point 5:The paper is well organized, but I would suggest improving the chapter structure a bit, starting with chapter 5, as follows:

  1. Discussion - I propose to start this chapter with the text in L. 338-342 and integrate it with the text in L. 349-385 (consequently, subsection 5.1. will disappear).
  2. Conclusions - I propose to begin with the text in L. 343-347 and then continue with 6.1. Theoretical implications, 6.2. Management imp., 6.3. Limitations & ... (you can also drop the separation of these subsections).

The paper is well written, I recommend it for publication after the authors have made these minor corrections.

Response 5:

       Thank you for your valuable advice. We are glad to adopt your suggestion and change the structure of the article to the following form:

5. Discussions

Analysing users' network behaviours and identifying users' preferences are undoubtedly the key to improve the performance of enterprises and e-commerce platforms. This study was a research endeavor to unearth the role of product price in influence of the CISB, including PL and PPD. We provided experimental evidence from the consumers’ behaviours on B2C e-commerce platforms regarding the PL and PPD on CISB. And our findings unearthed that the product category (durable and consumable) have a significant moderating effect on the relationship between PL, PPD and CISB.

Specifically,in the online shopping environment, the PL of durable goods has a significant positive impact on CISB. The more expensive the durable is, the higher the breadth and depth the consumer’s search effort is predicted to be. Because consumers have a higher perceived risk for higher-value, longer-lasting durable goods, more information searching behaviour is beneficial as a way to reduce consumers’ perceived risk and to obtain higher economic benefits. Conversely, PL was found to have a negative impact on the CISB of consumable goods. Product category (durable and consumable) was also found to have a significant moderating effect on the relationship between PL and CISB. It offers different conclusions from those put forward in existing literature [33, 45], that consumers will invest more effort searching for information about high-priced products. In this study, we found that CISB increases with the increase of product PL for durable goods. However, for consumable goods, which are characterized by a lower price and a higher frequency of purchase, consumers were shown to search for more information to obtain higher economic benefits for their frequent purchase behaviour. These conclusions are notable because of their consistency with information economics theory.

Furthermore, In the online shopping environment, PPD has a significant positive impact on CISB, and the product category has a significant moderating effect on the relationship between PPD and CISB. When consumers purchase consumables, their PPD has a more significant impact on the depth of CISB. Because consumables are purchased more frequently, higher degrees of PPD encourage consumers to make more trade-offs and comparisons on the price, discounts and preferential information about the products in order to ascertain the highest benefit from making the purchase. Different from the conclusions in other previous studies [11, 14, 15], in the online shopping environment, customers faced with a high level of PPD tend to prefer trusted brands or stores, and give up searching for more information and price comparisons. We found that, in the B2C online shopping environment, PPD has a positive impact on CISB. Therefore, we infer that on the B2C platform, consumers’ trust for retailers has improved, and they pay more attention to the cost performance of homogeneous products.

The third point is an inference. It can be found from the descriptive statistics of product data in Table 2, the depth and breadth of the search for information about refrigerators and tissues are much higher than for other products. We can easily understand that the PL of refrigerators is much higher than other products, but the phenomenon of tissues is unclear. We inspected the original consumer behavior data and found that consumers purchased tissues much more frequently than other products. So we speculate that if consumers need to buy certain consumables frequently, their search behavior for this product will be high [46].

6. Conclusions

This study answered how PL and PPD affect CISB synthetically. And we determined the impact direction of PPD on CISB in B2C e-commerce platform. In general, the research objectives were satisfactorily obtained, and our results had some implications and limitations

6.1. Theoretical implications

The results of study included some important theoretical implications in that, we prove that the PL and PPD of products affect the CISB at the same time, the PL affects the CISB through the degree of perceived risk, as more information search behavior can reduce the perceived risk [10]; and the positive impact of price dispersion on CISB is mainly due to the perceived potential benefits of more information search behavior [11, 12, 13]. In addition, our findings proved that consumers have different purchase decision-making thinking and paths for different categories of products from the perspective of consumers' information search behavior before purchase [38]. What’s more, our research provides a new idea for user behavior research based on large-scale e-commerce data.

6.2. Management Implications

This study has several important implications for marketing practice as well, by providing new insights into CISB for products of different PL and PPD. Our findings suggest that the characteristics of durable goods, which tend to have a higher PL, or consumable goods, or those which are purchased with higher frequency, encourage consumers to search for more information about these products. Therefore, we recommend marketing practitioners to provide consumers with abundant and varying product information to satisfy their need for such comprehensive levels of pre-purchase information. We also found that CISB is relatively lower for products characterized by moderate prices or a low PPD. Therefore, we recommend enterprises selling these goods to increase the intensity of advertising while ensuring consistently high product quality, to keep consumers who need to buy such products from looking to external sources for more knowledge about competing products. This strategy can help customers save time and energy in their search for regular use products, and help corresponding enterprises to sell more of their products, and faster.

6.3. Limitations and Future directions

In view of the limitations of network data characteristics and research methods, this article has the following limitations: 1) CISB is affected by many factors, such as individual product knowledge, search attitude and time pressure, the number of alternatives, brand differences, complexity and symbolism of the product [17]. This study only considers the influence of price as a factor on CISB. 2) This study adopts an empirical research method; based on information economics theory, the research results can be supplemented and improved by combining consumer information search models, such as optimal stop model, sequential search model [47], fixed search model [48], or others. 3) This study proved price factors significantly affect consumers’ search behavior before purchase and purchase decision, and future research could examine the association between price-sensitivity and consumer characteristics which can help to make targeted price promotions and effective personalized recommendation [49].
